# Design Principles of the Hippocampal Cognitive Map

**Kimberly L. Stachenfeld**[1]**, Matthew M. Botvinick**[1]**, and Samuel J. Gershman**[2]
[1]Princeton Neuroscience Institute and Department of Psychology, Princeton University
[2]Department of Brain and Cognitive Sciences, Massachusetts Institute of Technology
kls4@princeton.edu, matthewb@princeton.edu, sjgershm@mit.edu

## Abstract

Hippocampal place fields have been shown to reflect behaviorally relevant aspects of space. For instance, place fields tend to be skewed along commonly traveled directions, they cluster around rewarded locations, and they are constrained by the geometric structure of the environment. We hypothesize a set of design principles for the hippocampal cognitive map that explain how place fields represent space in a way that facilitates navigation and reinforcement learning. In particular, we suggest that place fields encode not just information about the current location, but also predictions about future locations under the current transition distribution. Under this model, a variety of place field phenomena arise naturally from the structure of rewards, barriers, and directional biases as reflected in the transition policy. Furthermore, we demonstrate that this representation of space can support efficient reinforcement learning. We also propose that grid cells compute the eigendecomposition of place fields in part because is useful for segmenting an enclosure along natural boundaries. When applied recursively, this segmentation can be used to discover a hierarchical decomposition of space. Thus, grid cells might be involved in computing subgoals for hierarchical reinforcement learning.

## 1 Introduction

A cognitive map, as originally conceived by Tolman [46], is a geometric representation of the environment that can support sophisticated navigational behavior. Tolman was led to this hypothesis by the observation that rats can acquire knowledge about the spatial structure of a maze even in the absence of direct reinforcement (*latent learning*; [46]). Subsequent work has sought to formalize the representational content of the cognitive map [13], the algorithms that operate on it [33, 35], and its neural implementation [34, 27]. Much of this work was galvanized by the discovery of *place cells* in the hippocampus [34], which selectively respond when an animal is in a particular location, thus supporting the notion that the brain contains an explicit map of space. The later discovery of *grid cells* in the entorhinal cortex [16], which respond periodically over the entire environment, indicated a possible neural substrate for encoding metric information about space.

Metric information is very useful if one considers the problem of spatial navigation to be computing the shortest path from a starting point to a goal. A mechanism that accumulates a record of displacements can easily compute the shortest path back to the origin, a technique known as *path integration*. Considerable empirical evidence supports the idea that animals use this technique for navigation [13]. Many authors have proposed theories of how grid cells and place cells can be used to carry out the necessary computations [27].

However, the navigational problems faced by humans and animals are inextricably tied to the more general problem of reward maximization, which cannot be reduced to the problem of finding the shortest path between two points. This raises the question: does the brain employ the same machinery for spatial navigation and reinforcement learning (RL)? A number of authors have suggested how RL mechanisms can support spatial learning, where spatial representations (e.g., place cells or

grid cells), serve as the input to the learning system [11, 15]. In contrast to the view that spatial representation is extrinsic to the RL system, we pursue the idea that the brain's spatial representations are designed to support RL. In particular, we show how spatial representations resembling place cells and grid cells emerge as the solution to the problem of optimizing spatial representation in the service of RL.

We first review the formal definition of the RL problem, along with several algorithmic solutions. Special attention is paid to the *successor representation* (SR) [6], which enables a computationally convenient decomposition of value functions. We then show how the successor representation naturally comes to represent place cells when applied to spatial domains. The eigendecomposition of the successor representation reveals properties of an environment's spectral graph structure, which is particularly useful for discovering hierarchical decompositions of space. We demonstrate that the eigenvectors resemble grid cells, and suggest that one function of the entorhinal cortex may be to encode a compressed representation of space that aids hierarchical RL [3].

## 2 The reinforcement learning problem

Here we consider the problem of RL in a Markov decision process, which consists of the following elements: a set of states $\mathcal{S}$, a set of actions $\mathcal{A}$, a transition distribution $P(s'|s,a)$ specifying the probability of transitioning to state $s' \in \mathcal{S}$ from state $s \in \mathcal{S}$ after taking action $a \in \mathcal{A}$, a reward function $R(s)$ specifying the expected reward in state $s$, and a discount factor $\gamma \in [0,1]$. An agent chooses actions according to a policy $\pi(a|s)$ and collects rewards as it moves through the state space. The standard RL problem is to choose a policy that maximizes the *value* (expected discounted future return), $V(s) = \mathbb{E}_\pi \left[ \sum_{t=0}^\infty \gamma^t R(s_t) \mid s_0 = s \right]$. Our focus here is on policy evaluation (computing $V$). In our simulations we feed the agent the optimal policy; in the Supplementary Materials we discuss algorithms for policy improvement. To simplify notation, we assume implicit dependence on $\pi$ and define the state transition matrix $T$, where $T(s,s') = \sum_a \pi(a|s)P(s'|s,a)$.

Most work on RL has focused on two classes of algorithms for policy evaluation: "model-free" algorithms that estimate $V$ directly from sample paths, and "model-based" algorithms that estimate $T$ and $R$ from sample paths and then compute $V$ by some form of dynamic programming or tree search [44, 5]. However, there exists a third class that has received less attention. As shown by Dayan [6], the value function can be decomposed into the inner product of the reward function with the SR, denoted by $M$:

$$V(s) = \sum_{s'} M(s,s')R(s'), \qquad M = (I - \gamma T)^{-1} \tag{1}$$

where $I$ denotes the identity matrix. The SR encodes the expected discounted future occupancy of state $s'$ along a trajectory initiated in state $s$:

$$M(s,s') = \mathbb{E} \left[ \sum_{t=0}^\infty \gamma^t \mathbb{I}\{s_t = s'\} \mid s_0 = s \right], \tag{2}$$

where $\mathbb{I}\{\cdot\} = 1$ if its argument is true, and 0 otherwise.

The SR obeys a recursion analogous to the Bellman equation for value functions:

$$M(s,j) = \mathbb{I}\{s = j\} + \gamma \sum_{s'} T(s,s')M(s',j). \tag{3}$$

This recursion can be harnessed to derive a temporal difference learning algorithm for incrementally updating an estimate $\hat{M}$ of the SR [6, 14]. After observing a transition $s \to s'$, the estimate is updated according to:

$$\hat{M}(s,j) \leftarrow \hat{M}(s,j) + \eta \left[ \mathbb{I}\{s = j\} + \gamma \hat{M}(s',j) - \hat{M}(s,j) \right], \tag{4}$$

where $\eta$ is a learning rate (unless specified otherwise, $\eta = 0.1$ in our simulations). The SR combines some of the advantages of model-free and model-based algorithms: like model-free algorithms, policy evaluation is computationally efficient, but at the same time the SR provides some of the same flexibility as model-based algorithms. As we illustrate later, an agent using the SR will be sensitive to distal changes in reward, whereas a model-free agent will be insensitive to these changes.

## 3 The successor representation and place cells

In this section, we explore the neural implications of using the SR for policy evaluation: if the brain encoded the SR, what would the receptive fields of the encoding population look like, and what

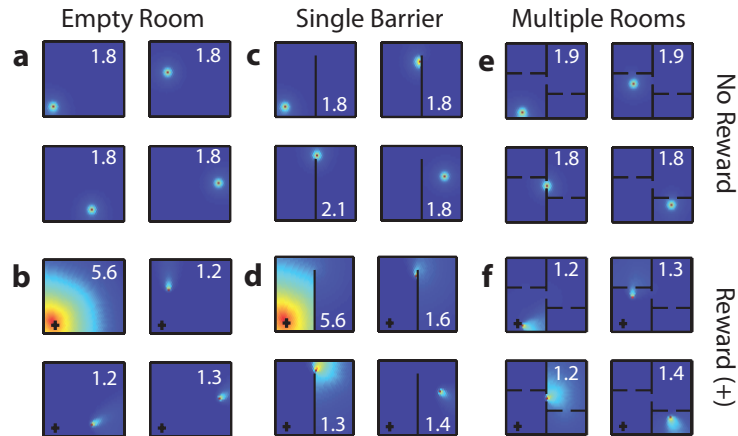

Figure 1: **SR place fields**. Top two rows show place fields without reward, bottom two show retrospective place fields with reward (marked by +). Maximum firing rate (a.u.) indicated for each plot. (a, b) Empty room. (c, d) Single barrier. (e, f) Multiple rooms.

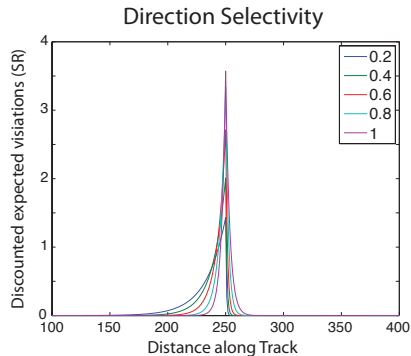

Figure 2: **Direction selectivity along a track.** Direction selectivity arises in SR place fields when the probability $p_\rightarrow$ of transitioning in the preferred left-to-right direction along a linear track is greater than the probability $p_\leftarrow$ of transitioning in the non-preferred direction. The legend shows the ratio of $p_\leftarrow$ to $p_\rightarrow$ for each simulation.

would the population look like at any point in time? This question is most easily addressed in spatial domains, where states index spatial locations (see Supplementary Materials for simulation details).

For an open field with uniformly distributed rewards we assume a random walk policy, and the resulting SR for a particular location is an approximately symmetric, gradually decaying halo around that location (Fig. 1a)—the canonical description of a hippocampal place cell. In order for the population to encode the expected visitations to each state in the domain from the current starting state (i.e. a row of $M$), each receptive field corresponds to a column of the SR matrix. This allows the current state's value to be computed by taking the dot product of its population vector with the reward vector. The receptive field (i.e. column of $M$) will encode the discounted expected number of times that state was visited for each starting state, and will therefore skew in the direction of the states that likely preceded the current state.

More interesting predictions can be made when we examine the effects of obstacles and direction preference that shape the transition structure. For instance, when barriers are inserted into the environment, the probability of transitioning across these obstacles will go to zero. SR place fields are therefore constrained by environmental geometry, and the receptive field will be discontinuous across barriers (Fig. 1c,e). Consistent with this idea, experiments have shown that place fields become distorted around barriers [32, 40]. When an animal has been trained to travel in a preferred direction along a linear track, we expect the response of place fields to become skewed opposite the direction of travel (Fig. 2), a result that has been observed experimentally [28, 29].

Another way to alter the transition policy is by introducing a goal, which induces a tendency to move in the direction that maximizes reward. Under these conditions, we expect firing fields centered near rewarded locations to expand to include the surrounding locations and to increase their firing rate, as has been observed experimentally [10, 21]. Meanwhile, we expect the majority of place fields

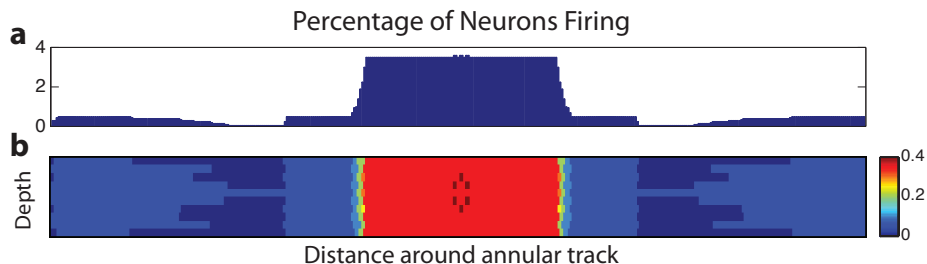

**Figure 3:** **Reward clustering in annular maze**. (a) Histogram of number of cells firing above baseline at each displacement around an annular track. (b) Heat map of number of firing cells at each location on unwrapped annular maze. Reward is centered on track. Baseline firing rate set to 10% maximum.

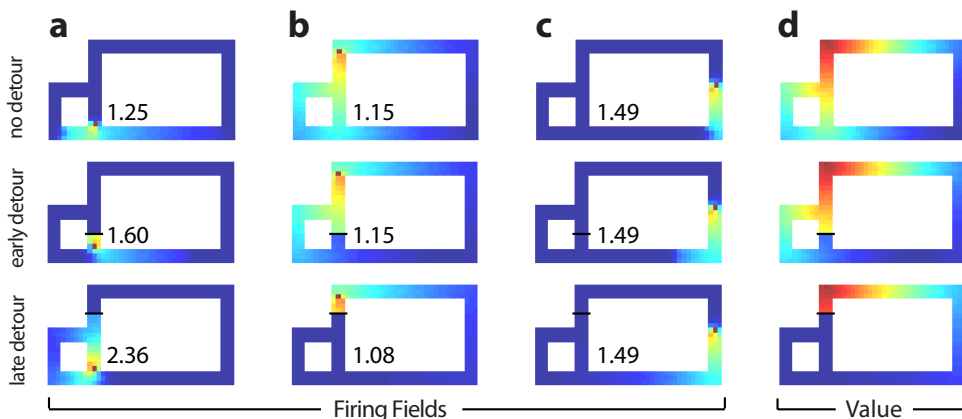

**Figure 4:** **Tolman detour task**. The starting location is at the bottom of the maze where the three paths meet, and the reward is at the top. Barriers are shown as black horizontal lines. Three conditions are shown: No detour, early detour, and late detour. (a, b, c) SR place fields centered near and far from detours. Maximum firing rate (a.u.) indicated by each plot. (d) Value function.

that encode non-rewarded states to skew slightly away from the reward. Under certain settings for what firing rate constitutes baseline (see Supplementary Materials), the spread of the rewarded locations' fields compensates for the skew of surrounding fields away from the reward, and we observe "clustering" around rewarded locations (Fig. 3), as has been observed experimentally in the annular water maze task [18]. This parameterization sensitivity may explain why goal-related firing is not observed in all tasks [25, 24, 41].

As another illustration of the model's response to barriers, we simulated place fields in a version of the Tolman detour task [46], as described in [1]. Rats are trained to move from the start to the rewarded location. At some point, an "early" or a "late" transparent barrier is placed in the maze so that the rat must take a detour. For the early barrier, a short detour is available, and for the later barrier, the only detour is a longer one. Place fields near the detour are more strongly affected than places far away from the detour (Fig. 4a,b,c), consistent with experimental findings [1]. Fig. 4d shows the value function in each of these detour conditions.

## 4 Behavioral predictions: distance estimation and latent learning

In this section, we examine some of the behavioral consequences of using the SR for RL. We first show that the SR anticipates biases in distance estimation induced by semi-permeable boundaries. We then explore the ability of the SR to support latent learning in contextual fear conditioning.

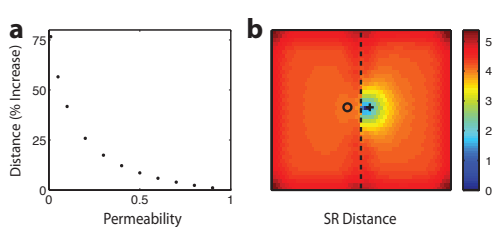

Figure 5: **Distance estimates.** (a) Increase in the perceived distance between two points on opposite sides of a semipermeable boundary (marked with + and ∘ in 5b) as a function of barrier permeability. (b) Perceived distance between destination (market with +) and all other locations in the space (barrier permeability = 0.05).

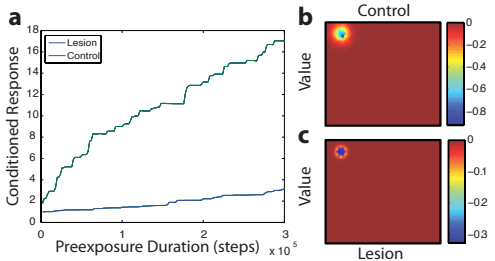

Figure 6: **Context preexposure facilitation effect.** (a) Simulated conditioned response (CR) to the context following one-trial contextual fear conditioning, shown as a function of preexposure duration. The CR was approximated as the negative value summed over the environment. The "Lesion" corresponds to agents with hippocampal damage, simulated by setting the SR learning rate to 0.01. The "Control" group has a learning rate of 0.1. (b) value for a single location after preexposure in a control agent. (c) same as (b) in a lesioned agent.

Stevens and Coupe [43] reported that people overestimated the distance between two locations when they were separated by a boundary (e.g., a state or country line). This bias was hypothesized to arise from a hierarchical organization of space (see also [17]). We show (Fig. 5) how distance estimates (using the Euclidean distance between SR state representations, $\sqrt{(M(s') - M(s))^2}$, as a proxy for the perceived distance between $s$ and $s'$) between points in different regions of the environment are altered when an enclosure is divided by a soft (semi-permeable) boundary. We see that as the permeability of the barrier decreases (making the boundary harder), the percent increase in perceived distance between locations increases without bound. This gives rise to a discontinuity in perceived travel time at the soft boundary. Interestingly, the hippocampus is directly involved in distance estimation [31], suggesting the hippocampal cognitive map as a neural substrate for distance biases (although a direct link has yet to be established).

The *context preexposure facilitation effect* refers to the finding that placing an animal inside a conditioning chamber prior to shocking it facilitates the acquisition of contextual fear [9]. In essence, this is a form of latent learning [46]. The facilitation effect is thought to arise from the development of a conjunctive representation of the context in the hippocampus, though areas outside the hippocampus may also develop a conjunctive representation in the absence of the hippocampus, albeit less efficiently [48]. The SR provides a somewhat different interpretation: over the course of preexposure, the hippocampus develops a *predictive* representation of the context, such that subsequent learning is rapidly propagated across space. Fig. 6 shows a simulation of this process and how it accounts for the facilitation effect. We simulated hippocampal lesions by reducing the SR learning rate from 0.1 to 0.01, resulting in a more punctate SR following preexposure and a reduced facilitation effect.

## 5 Eigendecomposition of the successor representation: hierarchical decomposition and grid cells

Reinforcement learning and navigation can often be made more efficient by decomposing the environment hierarchically. For example, the options framework [45] utilizes a set of subgoals to divide and conquer a complex learning environment. Recent experimental work suggests that the brain may exploit a similar strategy [3, 36, 8]. A key problem, however, is discovering useful subgoals; while progress has been made on this problem in machine learning, we still know very little about how the brain solves it (but see [37]). In this section, we show how the eigendecomposition of the SR can be used to discover subgoals. The resulting eigenvectors strikingly resemble grid cells observed in entorhinal cortex.

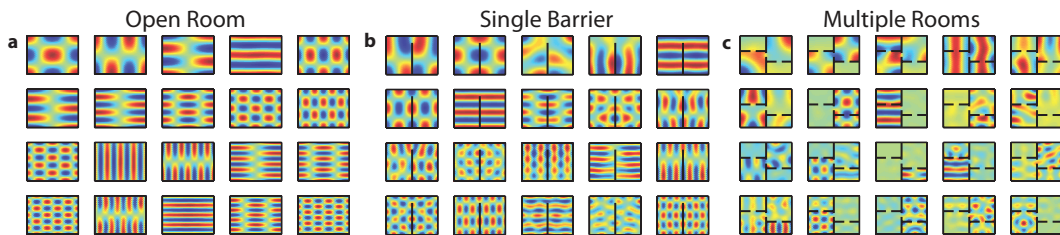

Figure 7: **Eigendecomposition of the SR**. Each panel shows the same 20 eigenvectors randomly sampled from the top 100 (excluding the constant first eigenvector) for the environmental geometries shown in Fig. 1 (no reward). (a) Empty room. (b) Single barrier. (c) Multiple rooms.

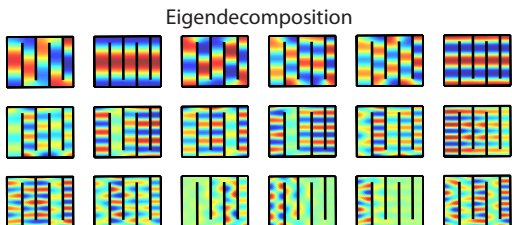

Figure 8: **Eigendecomposition of the SR in a hairpin maze**. Since the walls of the maze effectively elongate a dimension of travel (the track of the maze), the low frequency eigenvectors resemble one-dimensional sinusoids that have been folded to match the space. Meanwhile, the low frequency eigenvectors exhibit the compartmentalization shown by [7].

A number of authors have used graph partitioning techniques to discover subgoals [30, 39]. These approaches cluster states according to their community membership (a community is defined as a highly interconnected set of nodes with relatively few outgoing edges). Transition points between communities (bottleneck states) are then used as subgoals. One important graph partitioning technique, used by [39] to find subgoals, is the normalized cuts algorithm [38], which recursively thresholds the second smallest eigenvector (the Fiedler vector) of the normalized graph Laplacian to obtain a graph partition. Given an undirected graph with symmetric weight matrix $W$, the graph Laplacian is given by $L = D - W$. The normalized graph Laplacian is given by $\mathcal{L} = I - D^{-1/2}WD^{-1/2}$, where $D$ is a diagonal degree matrix with $D(s,s) = \sum_{s'} W(s,s')$. When states are projected onto the second eigenvector, they are pulled along orthogonal dimensions according to their community membership. Locations in distinct regions but close in Euclidean distance – for instance, nearby points on opposite sides of a boundary – will be represented as distant in the eigenspace.

The normalized graph Laplacian is closely related to the SR [26]. Under a random walk policy, the transition matrix is given by $T = D^{-1}W$. If $\phi$ is an eigenvector of the random walk's graph Laplacian $I-T$, then $D^{1/2}\phi$ is an eigenvector of the normalized graph Laplacian. The corresponding eigenvector for the discounted Laplacian, $I - \gamma T$, is $\gamma\phi$. Since the matrix inverse preserves the eigenvectors, the normalized graph Laplacian has the same eigenvectors as the SR, $M = (I-\gamma T)^{-1}$, scaled by $\gamma D^{-1/2}$. These spectral eigenvectors can be approximated by slow feature analysis [42]. Applying hierarchical slow feature analysis to streams of simulated visual inputs produces feature representations that resemble hippocampal receptive fields [12].

A number of representative SR eigenvectors are shown in Fig. 7, for three different room topologies. The higher frequency eigenvectors display the latticing characteristic of grid cells [16]. The eigendecomposition is often discontinuous at barriers, and in many cases different rooms are represented by independent sinusoids. Fig. 8 shows the eigendecomposition for a hairpin maze. The eigenvectors resemble folded up one-dimensional sinusoids, and high frequency eigenvectors appear as repeating phase-locked "submaps" with firing selective to a subset of hallways, much like the grid cells observed by Derdikman and Moser [7].

In the multiple rooms environment, visual inspection reveals that the SR eigenvector with the second smallest eigenvalue (the Fiedler vector) divides the enclosure along the vertical barrier: the left half is almost entirely blue and the right half almost entirely red, with a smooth but steep transition at the doorway (Fig. 9a). As discussed above, this second eigenvector can therefore be used to segment the enclosure along the vertical boundary. Applying this segmentation recursively, as in the normalized cuts algorithm, produces a hierarchical decomposition of the environment (Figure

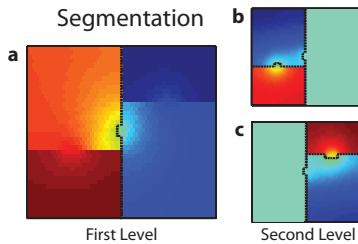

Segmentation

a

b

c

First Level    Second Level

Figure 9: **Segmentation using normalized cuts**. (a) The results of segmentation by thresholding the second eigenvector of the multiple rooms environment in Fig. 1. Dotted lines indicate the segment boundaries. (b, c) Eigenvector segmentation applied recursively to fully parse the enclosure into the four rooms.

9b,c). By identifying useful subgoals from the environmental topology, this decomposition can be exploited by hierarchical learning algorithms [3, 37].

One might reasonably question why the brain should represent high frequency eigenvectors (like grid cells) if only the low frequency eigenvectors are useful for hierarchical decomposition. Spectral features also serve as generally useful representations [26, 22], and high frequency components are important for representing detail in the value function. The progressive increase in grid cell spacing along the dorsal-ventral axis of the entorhinal cortex may function as a multi-scale representation that supports both fine and coarse detail [2].

## 6  Discussion

We have shown how many empirically observed properties of spatial representation in the brain, such as changes in place fields induced by manipulations of environmental geometry and reward, can be explained by a predictive representation of the environment. This predictive representation is intimately tied to the problem of RL: in a certain sense, it is the optimal representation of space for the purpose of computing value functions, since it reduces value computation to a simple matrix multiplication [6]. Moreover, this optimality principle is closely connected to ideas from manifold learning and spectral graph theory [26]. Our work thus sheds new computational light on Tolman's cognitive map [46].

Our work is connected to several lines of previous work. Most relevant is Gustafson and Daw [15], who showed how topologically-sensitive spatial representations recapitulate many aspects of place cells and grid cells that are difficult to reconcile with a purely Euclidean representation of space. They also showed how encoding topological structure greatly aids reinforcement learning in complex spatial environments. Earlier work by Foster and colleagues [11] also used place cells as features for RL, although the spatial representation did not explicitly encode topological structure. While these theoretical precedents highlight the importance of spatial representation, they leave open the deeper question of why particular representations are better than others. We showed that the SR naturally encodes topological structure in a format that enables efficient RL.

Spectral graph theory provides insight into the topological structure encoded by the SR. In particular, we showed that eigenvectors of the SR can be used to discover a hierarchical decomposition of the environment for use in hierarchical RL. These eigenvectors may also be useful as a representational basis for RL, encoding multi-scale spatial structure in the value function. Spectral analysis has frequently been invoked as a computational motivation for entorhinal grid cells (e.g., [23]). The fact that any function can be reconstructed by sums of sinusoids suggested that the entorhinal cortex implements a kind of Fourier transform of space, and that place cells are the result of reconstructing spatial signals from their spectral decomposition. Two problems face this interpretation. Fist, recent evidence suggests that the emergence of place cells does not depend on grid cell input [4, 47]. Second, and more importantly for our purposes, Fourier analysis is not the right mathematical tool when dealing with spatial representation in a topologically structured environment, since we do not expect functions to be smooth over boundaries in the environment. This is precisely the purpose of spectral graph theory: the eigenvectors of the graph Laplacian encode the smoothest approximation of a function that respects the graph topology [26].

Recent work has elucidated connections between models of episodic memory and the SR. Specifically, in [14] it was shown that the SR is closely related to the Temporal Context Model (TCM) of episodic memory [20]. The core idea of TCM is that items are bound to their temporal context (a running average of recently experienced items), and the currently active temporal context is used

to cue retrieval of other items, which in turn cause their temporal context to be retrieved. The SR can be seen as encoding a set of item-context associations. The connection to episodic memory is especially interesting given the crucial mnemonic role played by the hippocampus and entorhinal cortex in episodic memory. Howard and colleagues [19] have laid out a detailed mapping between TCM and the medial temporal lobe (including entorhinal and hippocampal regions).

An important question for future work concerns how biologically plausible mechanisms can implement the computations posited in our paper. We described a simple error-driven updating rule for learning the SR, and in the Supplementary Materials we derive a stochastic gradient learning rule that also uses a simple error-driven update. Considerable attention has been devoted to the implementation of error-driven learning rules in the brain, so we expect that these learning rules can be implemented in a biologically plausible manner.

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
