[Supplementary Material]

# Supplemental Information for Design Principles of the Hippocampal Cognitive Map

**Kimberly L. Stachenfeld**
Princeton Neuroscience Institute and Department of Psychology
Princeton University
kls4@princeton.edu

**Matthew M. Botvinick**
Princeton Neuroscience Institute and Department of Psychology
Princeton University
matthewb@princeton.edu

**Samuel J. Gershman**
Department of Brain and Cognitive Sciences
Massachusetts Institute of Technology
sjgershm@mit.edu

## 1 The learning algorithm

The SR-based critic learns an estimate of the value function, using the SR as its feature representation. Unlike standard actor-critic methods, the critic does not use reward-based temporal difference errors to update its value estimate; instead, it relies on the fact that the value function is given by $V(s) = \sum_{s'} M(s, s')R(s')$, where $M$ is the successor representation and $R$ is the expected reward in each state. Thus, learning $M$ and $R$ is sufficient to construct an estimate of $V$.

Learning $M$ can be accomplished using temporal difference learning, as described in the main text. Learning $R$ can also be accomplished using a simple delta rule:

$$\hat{R}(s) \leftarrow \hat{R}(s) + \eta_r \left[ r - \hat{R}(s) \right], \tag{1}$$

where $\eta_r$ is a learning rate and $r$ is the reward observed upon visiting $s$.

Even though the critic does not learn using reward-based temporal difference errors, it is still straightforward to compute them given an estimate of the value function, after observing transition $s \to s'$:

$$\delta = r + \gamma \hat{V}(s') - \hat{V}(s), \tag{2}$$

where $\hat{V}(s) = \sum_{s,s'} \hat{M}(s, s')\hat{R}(s')$. This error signal can be used to update the actor.

In particular, consider an actor who chooses action $a$ in state $s$ according to a softmax policy:

$$\pi(a|s) = \frac{\exp\{\beta U(s, a)\}}{\exp\{\beta \sum_{a'} U(s, a')\}}, \tag{3}$$

where $\beta$ is an inverse temperature parameter. The state action weights $U$ are updated according to:

$$U(s, a) \leftarrow U(s, a) + \eta_u \delta, \tag{4}$$

where $\eta_u$ is a learning rate.

## 2 Learning the eigendecomposition through stochastic gradient descent

We are interested in computing the $K$ largest eigenvectors of the discounted successor representation. This problem is equivalent to choosing the embedding matrix $Y$ that minimizes the following cost function:

$$C(Y) = \sum_s \sum_{s'} E(y_s, y_{s'})T(s, s'), \tag{5}$$

where $y_s \in \mathbb{R}^K$ is the embedding of state $s$ and $E(y_s, y_{s'}) = \gamma ||y_s - y_{s'}||^2$. Intuitively, this cost function favors embeddings that are similar for states that are likely to be visited sequentially. The optimal embeddings can be viewed as the solution to a kernel PCA problem with $M$ as the kernel.

The cost function can also be viewed as an expectation under the Markov chain induced by $\mathbf{T}$:

$$C(Y) = \mathbb{E}_{s' \sim T(s, \cdot)} \left[ E(y_s, y_{s'}) \right]. \tag{6}$$

We can therefore minimize the cost function online by sampling transitions $(s \to s')$ from the Markov chain and stochastically following the gradient:

$$y_i^{n+1} = y_i^n - \alpha^n \nabla_{y_i} E(y_s^n, y_{s'}^n), \tag{7}$$

where $\alpha^n$ is the step-size (learning rate) at time $n$, and the gradient is given by:

$$\nabla_{y_i} E(y_s^n, y_{s'}^n) = 2\eta\gamma \frac{y_s^n - y_{s'}^n}{||y_s^n - y_{s'}^n||}, \tag{8}$$

$$\eta = \begin{cases} 1 & \text{if } i = s \\ -1 & \text{if } i = s' \\ 0 & \text{otherwise.} \end{cases} \tag{9}$$

## 3 Simulation methods

### 3.1 Spatial representations

We began by discretizing the spatial domain. Four different models of space were used over our simulations.

1. *Rectangular grid.* (Fig. 1, 5–9). Here we discretized the spatial domain to create a $40 \times 40$ hexagonal grid. Corresponding to this grid is a graph containing $N = 1600$ states, in which transitions were possible to six equidistant adjacent states (Supp. Fig. 1a).

2. *Tolman Detour Maze (Fig. 4).* To produce a discretized Tolman detour maze, we started a rectangular $42 \times 21$ rectangular grid in which transitions were once again possible along cardinal directions and diagonals. We then removed all states that fell within the following rectangles on the grid: $[1, 9] \times [13, 21]$, $[4, 6] \times [4, 6]$, and $[10, 39] \times [4, 18]$. The resulting grid was a Tolman detour maze with a track of width 3 (Supp. Fig. 1b).

3. *Annular Maze (Fig. 3).* To produce a annular maze, we simply discretized a $10 \times 120$ rectangular grid and knitted together the edges to make the track circular. While this neglects some of the subleties of the curvature, the topology is basically the same.

4. *1D Linear Track (Fig. 2).* To examine direction selectivity over a linear track, we used a chain of 500 states connected in a chain.

Within the discretized space, the animal's transitions can be considered a Markov Process over the graph in which transition probabilities are proportional to the edge weights. We first computed the adjacency weight matrix $W$ of this undirected graph. This adjacency matrix is an $N \times N$ matrix in which $W_{ss'}$ is set to a 1 if and only if a transition from state $s$ to state $s'$ is possible ($N$ is the number of states).

### 3.2 Obstacles, reward, and bias

To model insurmountable obstacles and barriers in the environment, we performed "cuts" along the grid, severing the transitions between states $s$ and $s'$ on opposite sides of a barrier by replacing $W_{ss'}$ with 0 (Fig. 1c–f, 4, 7,

Figure 1: **Discretization** (a) Illustration of the discretization of the animal's enclosure. As shown, transitions are possible in cardinal directions and along diagonals. We used a $40 \times 40$ grid for our simulations, but here show a $10 \times 10$ grid for better viewing resolution. (b) Discretization used for Tolman detour maze. Dimensions of the bounding rectangle are $42 \times 21$.

8). To model soft barriers that are surmountable with some degree of difficulty or uncertainty, which we refer to as "semipermeable" barriers, we set $W_{ss'}$ instead equal to a permeability value $p$ between 0 and 1 (Fig. 5). To capture directional preferences, we make the weight matrix asymmetric such that an anti-preferred transition from $s$ to $s'$ is associated with weight $W_{ss'} < W_{s's} = 1$ (Fig. 2). Since, in the process of calculating the transition probability matrix, we normalize rows so that they sum to 1, the absolute values in the adjacency matrix have no real meaning; rather, it is their relative values that make a difference.

From the adjacency information provided by this grid, we compute the transition probability matrix $T$. Each element $T_{ss'}$ in the transition probability matrix corresponds to the probability of transitioning from state $s$ to $s'$. In the simplest case, the transition policy is simply a random walk, and the probability of transitioning from state $i$ to an adjacent state is simply $1/N_{\text{adj}}$, where $N_{\text{adj}} = \sum_{s'=1}^{N} W_{ss'}$ is the number of adjacent nodes. In other words, we normalize the rows of the adjacency matrix so that they sum to 1.

In more complex cases, the transition policy is not simply a random walk and is instead influenced by behaviorally relevant features such as reward or punishment. To implement this, we assigned nonzero reward to certain locations and used policy iteration to determine the optimal transition policy and state valuation. We assume that the rat has some nonzero tendency to explore (or at least to show some random deviations from the optimal path) even when the locations of some rewards are known, and therefore used a softmax transition policy (discussed in Supplemental Section 1).

### 3.3 Computing the successor representation

Now that we have our transition matrix, we can compute the discounted successor representation, $M$. The successor representation can be learned from TD, as described in Supplemental Section 1, or by applying the rule for calculating the sum of a geometric series of matrices. For most of our simulations, we implemented the latter and compute the SR using $M = \sum_{t=1}^{\infty} \gamma^t T^t = (I - \gamma T)^{-1}$ (Fig. 1–5, 7–9). For Fig. 6, in which we examine training time dependent effects, we use the temporal differences algorithm to produce a profile of how place fields might look during formation (described in Supplemental Section 1).

The $s^{\text{th}}$ column of the successor representation matrix encodes the discounted expected number of visitations to state $s$ from every other starting in state in the graph, given a particular transition policy and discount $\gamma$. These columns will make up our receptive fields. The population vector corresponding to some state $s$ will therefore be the $s^{\text{th}}$ row of the successor representation matrix — a slice through all receptive field columns at a given state. Thus, the population encodes the discounted expected number of visitations to each other state when starting in state $s$ under the transition policy and discount. To visualize the $s^{\text{th}}$ place field, we reshape the SR column so that the successor states are aligned to their $x, y$ coordinates in the spatial domain.

From the SR, we could compute the value $V$ over the state space by taking the product $V = MR$. This vector was reshaped and plotted to fit the space in Fig. 4d, 6. We also displayed the distance between states $s$ and $s'$ in the successor representation for Fig. 5 which required simply computing the Euclidean distance between SR column

| | | | | |
|---|---|---|---|---|
| 10 | 11 | 13 | 19 | 20 |
| 21 | 29 | 32 | 33 | 39 |
| 42 | 50 | 53 | 54 | 55 |
| 59 | 64 | 67 | 71 | 72 |
| 77 | 78 | 84 | 86 | 92 |

| | | | | | |
|---|---|---|---|---|---|
| 10 | 11 | 13 | 19 | 20 | 21 |
| 29 | 32 | 33 | 39 | 42 | 50 |
| 53 | 54 | 55 | 59 | 64 | 67 |
| 71 | 72 | 77 | 78 | 84 | 86 |

Table 1: **Eigenvector indexing.** The eigenvectors corresponding to each panel in Figure 7 (left) and Figure 8 (right).

vectors corresponding to the two states, $\sqrt{(M_{s'} - M_s)^2}$. To quantify reward clustering, we counted the number of fields that had an SR value above some baseline covering each location. Baseline was set to 10% of the population's maximum possible firing rate for Fig. 3.

### 3.4 Eigendecompositions

To generate grid cells, we computed the eigendecomposition of the successor representation. In any figure showing $k$ eigenvectors, we the same $k$ eigenvectors sampled from the 2nd through 100th lowest frequency eigenvectors. The eigenvectors selected are displayed in Supp. Table 1.

Since each row of the SR sums to $1 - \gamma$, the first eigenvector will always be a constant vector and is therefore uninteresting to us. In Section 5, we show that the 2nd eigenvector is useful for segmentation of an enclosure along natural boundaries. For simplicity, we threshold at zero, though more complicated thresholding schemes based on the histogram of the values in the eigenvector can be employed.

We then recursively compute eigendecompositions of the halved room in order to subdivide it further. Segmentation can be stopped when the second eigenvector meets a "smoothness criterion," which indicates that there are no natural boundaries dividing the enclosure. Smoothness arises because in an unobstructed room, the eigenvectors are discrete sinusoids with no discontinuities.

### 3.5 Parameters

All simulations were run using discount $\gamma = 0.98$ and an inverse temperature parameter $\beta = 2.5$. For simulations in which the SR was computed using TD (Fig. 5), learning rates of $\eta_m = 0.01$ and $\eta_m = 0.1$ were used. Increasing the discount $\gamma$ affects how far-sighted the animal will be. In the absence of reward, increasing $\gamma$ simply increases the width of the place field, since longer journeys matter more in computing the SR (Supp. Fig. 2a). We see increased skewing away from the rewarded location as $\gamma$ increases because states that occurred earlier in a walk towards the goal state are representing non-negligibly (Supp. Fig. 2b). We also

The softmax inverse temperature parameter modulates to what extent the animal relies on its optimal policy. Decreasing $\beta$ for softmax policy will push the animal's policy toward a random walk, and increasing $\beta$ will push the animal's policy toward the current optimal policy. A low-valued $\beta$ will therefore produce a nearly circular place field, and a high valued $\beta$ will produce a more narrow place field that extending in the direction of reward (Fig. 3).

The learning rate for the SR, $\eta_m$, as is suggested by the name, modulates the rate at which the SR is learned. A very low learning rate will cause the SR to converge very slowly. A higher learning rate will allow the SR to converge more quickly; however, too high a learning rate precludes convergence. Shown in Supp. Fig. 5 is the mean squared error between the SR computed from TD and the "true" SR computed from matrix inversion for different learning rates as function of training time.

To compute the percentage of cells firing at each location, we counted the cells with SR fields that were above some baseline at the location. Altering that baseline had a significant effect on the extent to which clustering could be observed around the rewarded location (Supp. Fig. 5). If the threshold is high, than the fact that the fields near the reward have much larger firing rates will compensate for the fact that more distant fields skew *away* from the reward,

Figure 2: **Effect of changing discount** Shown in a–c are the SR place fields for a random walk policy in a room containing a barrier with (a) $\gamma = 0.5$, (b) $\gamma = 0.95$. and (c) $\gamma = 0.995$. As the $\gamma$ increases, the place field grows to fill the left room. In the bottom row, we see the SR place fields when there exists reward in the same environment. We see that the extent to which place fields are skewed toward the rewarded location increases with $\gamma$. (d) $\gamma = 0.5$, (e) $\gamma = 0.95$. (f) $\gamma = 0.995$.

Figure 3: **Effect of changing discount** Shown above is the effect of varying the softmax inverse temperature parameter, $\beta$, on place fields. (a) $\beta = 0$. (b) $\beta = 2.5$. (c) $\beta = 10$.

and clustering can be observed. If the threshold is relatively low, the the skew away from the reward of the cells with lower overall firing rates is taken fully into account, and there is reduced firing near the reward.

Figure 4: **Effect of changing learning rate** Mean squared error between the SR learned from TD and the "true" SR, $M$ over time for a variety of learning rates.

# Percentage of Neurons Firing

Figure 5: **Effect of setting different baseline threshold on reward clustering**. The fraction of maximum firing rate that maximum was set to is given on the left.