[Reviews · NeurIPS 2014]

Submitted by Assigned_Reviewer_6

Beyond representing the position of an animal in a given environment, the activity of neurons in the hippocampus (areas CA1,3) is known to be influenced by a range of task-dependent factors, for instance the presence of a reward at a specific location in the environment; yet we don't fully understand how these representations emerge and what they are good for. The present paper proposes that these observations are a reflection of the circuit implementing a specific algorithm (using a successor representation, SR, initially proposed by Dayan in 1993) for learning state values for reinforcement learning; moreover it suggests that the representation in an upstream region (medial EC) may provide a basis for a hierarchical decomposition of space. Overall, some of the ideas put forward here are intriguing and potentially interesting for theoretical neuroscientists studying hippocampal coding, however the link to the neural data is relatively weak and the presentation of the material is difficult to follow in places.

Detailed comments

1. Content:
Since the algorithmic part of the paper is not new, the key contribution of this work is the link between the SR representation and the activity of neurons in the hippocampus. Unfortunately, this link between the two is not clear in several aspects:

a) it is never spelled out exactly how does the matrix M relate to the firing of the neurons in the corresponding hippocampal circuit.If there is a one-to-one map between firing rates and M(s,s'), how can a downstream circuit compute V(s)?
One would need to include at least a cartoon version of how the whole algorithm would work at the level of the circuit.

b) in SR the representation is a direct reflection of the geometry of the space on one side and the policy on the other.
It is the second component that is particularly important for explaining task-dependent effects on neural activity; still, in section 3 it is barely stated why policy corresponds to which experimental result (there is a general description of the different learning algorithms but it is never spelled out what exactly was used when). This is particularly important from the perspective of the dynamics of learning in the circuit: in an online setting, the policy changes in parallel to the state representation, which presumably could influence the final result (depending on the time constants for learning the different components).

c) On a more computational note, the parallel learning of policy and representation can actually slow down learning in a some settings, for instance when the rewards contingencies change (see Dayan 93). In a constantly changing environment, it is not clear to me why it would be a good idea to have a policy-dependent representation rather than just doing proper model-based RL. Presumably it is the ease of learning that makes such a solution still appealing; these issues should be at least mentioned in the discussion.

c) I find the link to the Stevens and Coupe behavioural results somewhat tenuous: in particular the notion of a boundary as defined there is slightly different than a simple spatial boundary (be it a 'semi-permeable' one); also I would not ascribe the processing of state boundaries to the hippocampus in particular; if nothing else, this bit needs to be justified more carefully.

d) the activity of grid cells in MEC looks nothing like the fields in fig 6. Although the responses are periodic, the geometry is different (hexagonal grid) and border cells have elongated fields, these are restricted to the actual border and do not show any periodicity; I am unaware of any experimental data corresponding to panel c. More generally, I am unaware of any experimental work showing reward-dependent changes in these responses, which would be expected should these neural responses correspond to an eigendecomposition of the SR.

Small technical note: the convolution with the Gaussian filter is used in the data analysis to smoothen out some of the noise due to limited sampling of the neural responses. I don't see a reason to use it here, and in fact smoothening the fields may prove misleading. Also, was this smoothing used consistently? (there does not seem to be any in the plots with boundaries)

2. Clarity: I find the structure of the paper is somewhat difficult to follow, in particular the fact that important technical details are lumped together in the supplements, often without even being referenced in the main text. Also some of the details are missing, making it impossible to know exactly what was done to obtain the results in individual figures.

3. Significance: once the link to the neural circuit is made explicit and the presentation is improved, this could be a very interesting paper for the hippocampal coding subfield.
Summary: The paper puts froward some interesting ideas on hippocampal representations as a substrate for RL unfortunately it is let down by limited link to experiments and poor presentation.

Submitted by Assigned_Reviewer_25

The paper „Design Principles of the Hippocampal Cognitive Map“ presents a new model that explains the emergence and use of place cells and grid cells in hippocampus as implementation of a successor representation (SR) of the states for subsequent RL learning.
The model nicely explains experimentally observed distortions in the shape of place fields as they are induced by barriers or direction travel. Grid cells are compared to the result of an eigenvector decomposition of the SR, which provides a hierarchical decomposition of the space according to functional barriers.

The explanatory power of this model is very convincing and so clear that I wonder why this relation between Hippocampal coding and SR was not shown earlier. To the best of my knowledge the work is original.

The paper is clearly structured, although the mathematical derivation of the model is very condensed. The supplemental information is a necessary part of the paper.

I am sure that this work will have a broad impact and is of great interest for the NIPS-community.

PS: Supplement p2: 3.1.: Please correct the numbers in the grid definition of the Tolman Detour task.
Summary: The paper „Descign Principles of the Hippocampal Cognitive Map“ presents a new model that explains the emergence and use of place cells and grid cells in hippocampus as implementation of a successor representation (SR) of the states for subsequent RL learning.
I am sure that this work will have a broad impact and is of great interest for the NIPS-community.

Submitted by Assigned_Reviewer_32

The main claim of the paper is that spatial representations emerge as a component of the solution to reinforcement learning (RL) problems, and therefore reflect the main attributes and desiderata of RL. Specifically, the authors argue that the major properties of place cells and grid cells arise from optimizing spatial representations for RL. For example, three basic properties of place cells, their skewness in commonly traveled directions, their clustering around rewarded locations, and their geometrically reflecting constraints, are argued to result from the above optimality principles. The main arguments of the paper are presented for the case of policy evaluation, although some discussion of policy improvements is presented in the appendix. The results are demonstrated through simulations of maze based environments with varying geometric structures and constraints.

The starting point for the paper is the so-called “successor representation” (SR) developed in [6], where each state is characterized by a vector encoding the discounted future expected occupancy of all other states. The value function is then computed as an inner product of this vector with the reward, and a TD-like algorithm is proposed for evaluating it. The SR representation is argued to play an intermediate role between model-based and model-free approaches. Through a series of maze simulations in different environments, the authors demonstrate that the basic features of place cells, alluded to above, arise. In addition to explanations of physiological properties of place cells, the authors also suggest some behavioral consequences. First, they argue that distance estimates arising from the SR, depending on the permeability level of boundaries, are consistent with perceived distances by humans. Additionally, they also suggest an explanation for context pre-exposure facilitation. Finally, they argue that grid cells emerge through an eigen-decomposition of place cells, and that this representation is beneficial for hierarchical RL.

This is a well written paper, reflecting a novel and original set of ideas, and leading to non-trivial explanations of observed phenomena, as well as predictions for future experiments. While based mostly on simulations, it has done a good job of convincing the reader of the correctness of the results. The paper offers a clear working hypothesis, which is original in the present context (as far as I can tell), and goes a long way towards establishing its relevance and explanatory power.

A few issues minor points:
Lines 195-200: This experimental setting and results in this paragraph are not clear to me, and require an improved explanation.
Line 214: The ‘distance’ measure introduced here is non-symmetric, so cannot be really referred to as distance (unless I am missing something here).
Line 269: The graph Laplacian is introduced based on a symmetric weight matric W. The authors do not explain what W is in the present context. Given that it is related to the previously introduced matrix M and its non-symmetric variant introduced in line 214, I don’t see how the symmetry issue is addressed.
Line 321: It was not clear to me how the hierarchical nature of representation in RL arises from this decomposition, and, specifically, how sub-goals are attained.
Line 261: The striking resemblance of the emerging grid cells to those observed in entorhinal cortex is noted. Could this statement be made more quantitative? For example, how does the nature of the grid (square, hexagonal, etc.) depend on the properties of the environment, if at all.
The present paper considers the effect of place cells on grid cells. However, it is well known that grid cells directly affect place cells. Can the authors envisage any benefit, within their framework, to such an influence?
Summary: A well written paper proposing a novel explanation for place cells (and to some extent grid cells) in the context of reinforcement learning. The paper leads to non-trivial explanations of observed phenomena, and to predictions for future experiments.
Author Feedback
Author rebuttal: We would like first to thank the reviewers for the opportunity to clarify some important points. Please note that we will be happy to work the replies below into the paper and supplement as recommended.

One question raised was that of how the matrix M relates to the firing of hippocampal neurons. As the reviewer inferred, we do assume a one-to-one mapping between elements of the SR and activity of individual neurons (although any monotonic, linear mapping would work equally well). On the question of how the function V = M*R might be computed: In fact, this can be accomplished by an extremely simple neural network. All that is required is a combination of additive and multiplicative synaptic interactions, both of which are observed ubiquitously in the CNS. We would be happy to provide a detailed description of the network required, either in this forum or in the supplement, as desired.

With regard to which learning algorithm is used in which simulation, please note that our paper explicitly states that, in all simulations, the optimal policies were simply provided rather than learned. The paper also states that the SRs were computed in closed form for all figures besides those for which training length effects were simulated (i.e. Figure 5, in which the SR was learned by TD). The purpose of the learning-algorithm descriptions is to show that the policies and SRs *can* be learned. Previously published work on the SR (including Peter Dayan’s original paper) allows us to state categorically that learning would have yielded precisely the same policies and representations as those involved in our simulations using either learning algorithm.

One reviewer questioned why it might be preferable to rely on SRs rather than to do proper model-based RL. In fact, we do not think of the SR as a substitute for model-based RL, but rather as a complement to it. Even if the SR is based on a policy quite different from one the agent must discover given a new situation, as long as the SR retains an adequate ‘exploratory’ flavor — that is, as long as the policy on which the SR is based retains a strong stochastic component — the product M*R will still be a very good ‘initial draft’ of the value function. This initial draft can then be fine-tuned through model-based RL. This two-part approach to model-based RL has been applied with dramatic success in recent work on large-scale problems, famously including the game of go (see, e.g., the work of David Silver and Rich Sutton on Dyna-2). Importantly, the robust usefulness of the SR is amplified in settings with a great deal of structure, like those that are the focus of our paper. We hesitate to delve into this side of the story in the present paper, but will endeavor to work it in if this is recommended.

Concerning the differences between the activity of MEC grid cells and the fields in figure 6: This is an artifact of having discretized with a rectangular-with-diagonal grid rather than triangular lattice grid. Eigenvectors of triangulated graphs in fact yield the classical hexagonal pattern, as do the eigenfunctions of an open field in the continuous analogue. We intend to provide updated figures using the triangular latticing, a move that is justified by the fact that the discretization is ultimately meant to approximate the continuous case.

We also want to add that although we primarily discuss a role for grid cells downstream of place fields, a more general interpretation might be that grid cells simply encode information in the functional basis ‘preferred’ by the neocortex. Our paper provides a direct explanation for why representation in the spectral basis facilitates segmentation, value representation, and compression. Normatively, it makes sense for the hippocampus to encode outgoing information in such a basis, and conversely, for the cortex to encode ingoing information in the same basis. As the paper notes, this idea is also consistent with available anatomical and experimental findings.

We agree that the boundaries studied by Stevens & Coupe are different in character from tangible, obstructing boundaries in an animal's immediate vicinity. However, we do not necessarily find it unlikely that those more abstract boundaries are incorporated into a cognitive spatial map in a similar, albeit more flexible/uncertain way, particularly during tasks specifically involving evaluating spatial relations. It is difficult to probe large scale cognitive maps with animal models, as we can really only probe cognitive maps during active (as opposed to imagined) navigation. As such, we are resorting to human behavioral findings that exist, and we believe a hippocampal representation of large scale cognitive maps is an experimental avenue worth pursuing in humans. Findings from Russell Epstein’s lab, for example, provide some encouragement for this perspective (e.g., Morgan et al., 2011).

With regards to the note about smoothing, we smoothed with a very small kernel, so the effect of the convolution on the place fields shown is quite minimal. Smoothing was undertaken for the sake of consistency with data analysis procedures, since fields shown experimentally have undergone the same type of smoothing we apply. However, we too were concerned that a large amount of smoothing would be misleading, and in fact diminishes the effects we sought to demonstrate (as smoothing over boundaries diminishes the effect boundaries have on place fields under our model). Additionally, since we are modeling the smooth underlying function of the place field, we agree that the smoothing step is arguably optional. All place field plots were smoothed with the same kernel, even those corresponding to environments with boundaries; this is evidence of just how small the effect of smoothing is on place fields.

Once again, we would like to thank the reviewers for their time and helpful comments. We hope the replies we have provided here will satisfy all concerns.